

**Internal tides off the Amazon shelf in the western tropical Atlantic: Analysis of**
**SWOT Cal/Val Mission Data**
Michel Tchilibou[1], Loren Carrere[1], Florent Lyard[2], Clément Ubelmann[3], Gérald Dibarboure[4],
Edward D. Zaron[5], and Brian K. Arbic[6]
[1] Collecte Localisation Satellites, 31520 Ramonville-Saint-Agne, France
[2] Université de Toulouse, LEGOS (CNES/CNRS/IRD/UT3), 31400 Toulouse, France
[3] Datlas, Grenoble, France
[4] Centre National d'Etude Spatiales, 31400, Toulouse, France
[5] Department of Civil and Environmental Engineering, Portland State University, Portland, OR 97207-0751, USA
[6] Department of Earth and Environmental Sciences, University of Michigan, Ann Arbor, MI, USA
Correspondence to: Michel Tchilibou (mtchilibou@groupcls.com)
**Abstract**
The Surface Water and Ocean Topography (SWOT) altimetry mission launched at the end of 2022 is
an opportunity to access ocean variability at scales down to 15-30 km and to better understand high-
frequency dynamic processes such as the internal tide (IT). This study characterizes the internal tides
off the Amazonian shelf in the tropical Atlantic; it is based on 2 km horizontally gridded observations
along the swaths of SWOT track 20 during the calibration/validation phase (Cal/Val, 1-day orbit) from
late March to early July 2023. Internal tide models for M2, S2 and N2 were first derived by harmonic
analysis of the sea level anomaly (SLA), then improved by performing a principal component analysis
(PCA) prior to harmonic analysis. The results compare very well with the high-resolution empirical tide
(HRET) internal tide model, the reference product for internal tide corrections in altimetry
observations. The coherent mode 1 and mode 2 can be distinguished in the internal tide model derived
from SWOT, while the higher modes with their strong SLA signature seem mostly in the incoherent
part. The PCA also gives an overview of the daily variability of the internal tide.
**Introduction**
The launch of the SWOT mission at the end of 2022 certainly marks a new phase in spatial altimetry.
SWOT is equipped with the KaRIn instrument, a Ka-band radar interferometer capable of measuring
the sea surface topography with unprecedented resolution. KaRIn consists of two antennae that take
2D measurements in two 50 km-wide bands separated by a 20 km gap covered by the conventional
nadir radar altimeter also carried by the mission. The accuracy of SWOT's instruments is such that
SWOT should be able to observe the ocean down to a spatial scale of 15-30 km (Morrow et al., 2019;
Dufau et al., 2016; Wang et al., 2019), thus, complementing our 2D view of the ocean with
Topex/Poseidon class nadir altimetry, which is limited to scales larger than 150 km (Chelton et al.,
2011; Ballarotta et al., 2019) along one-dimensional tracks rather than two-dimensional swaths. The
main oceanographic objective of the SWOT mission is to characterize mesoscale and sub-mesoscale
ocean circulation (Fu et al., 2012; Fu and Ubelmann, 2014). However, ocean processes at the scales
targeted by SWOT (150-15 km) encompass both "balanced" geostrophic motions, as well as surface
and internal inertia-gravity waves at tidal frequencies. The prediction of internal tides (IT) presents a
significant challenge to the useability of SWOT data, considering that the spatial scales of these waves



overlap with those of balanced motions. Conversely, the exploitation of SWOT data to study IT is an
opportunity for learning more about these waves and quantifying their impacts in the ocean.
Efforts have been made in recent years to map internal tide using conventional altimetry
observations. This was made possible by the fact that the internal tide has a SSH (Sea Surface height)
signature of the order of one to several centimeters (Chelton et al., 1998; Ray and Mitchum, 1997).
However, the coarse sampling in both space and time of conventional altimetry is a hindrance. To
derive spatially continuous high-resolution maps of the internal tide SSH from the sparse altimeter
sampling, Dushaw (2015), Zhao et al. (2019) and Zaron (2019) used least-squares techniques to fit
kinematic wave solutions to nadir altimetry. Ubelmann et al., 2022 proposed jointly estimating internal
tides and mesoscale eddies to produce 2D maps of internal tides from conventional altimetry
observations. The advent of SWOT is an opportunity to validate these internal tide maps using direct
2D observations of the ocean.
Following the linear theory of ocean vertical modes, internal tides can be decomposed as a sum of
orthogonal baroclinic modes (Gill, 1982; Kelly et al., 2016). The first modes (mode 1 and mode 2)
propagate over hundreds or even thousands of kilometers. Higher modes have much shorter
wavelengths and are likely to dissipate close to the internal tide generation site, due to their low group
velocity and high shear (St Laurent and Garrett, 2002; Vic et al.,2019), and could barely be observed in
classical nadir altimetry observations. SWOT's high resolution is thus an opportunity to better observe
these higher modes.  In practice, the internal tide is separated into the so-called coherent and
incoherent internal tides. The coherent internal tide is the part of the internal tide which remains
phase-locked with the generating barotropic tide over an arbitrary period and are easily obtained by
harmonic analysis over the targeted period, as harmonic analysis will only retain local amplitude and
phase locked contributions. Consequently, the residual that escapes harmonic analysis constitutes the
incoherent internal tide.  The amplitude, phase, and trajectory of incoherent internal tide results from
refraction, reflection, and advection of internal tide by the ocean background circulation including
eddies, currents, and stratification (Ponte and Klein, 2015; Nelson et al.,2019; Buijsman et al., 2017;
Dunphy et al., 2017; Dunphy and Lamb, 2014; Duda et al., 2018; Savage et al., 2020; Barbot et al.,
2021). As SWOT can capture both tides and eddies surface signatures, it provides an opportunity to
investigate their interaction, to get insight of the incoherence of internal tide and, hopefully, to take
up the challenge of their separability.
Like the barotropic tide, the internal tide is a mixture of long- and short-period waves, among
which the main astronomical tides, such as the diurnal waves (O1, K1, P1) and the semi-diurnal waves
(M2, S2, N2, K2). Due to the low repetitiveness of altimetry satellites, short tidal periods are aliased to
longer periods (Le Provost, 2001).  The M2 tide, for example, is aliased to 62.11 days for the
TOPEX/Jason 10-day orbit (9.92 days precisely). With SWOT sampling, M2 is aliased to 66.02 days or
12.35 days (Table 1), depending on whether we consider the 21-day final science orbit or the 1-day
calibration/validation (Cal/Val) orbit (0.99343 days exactly).  Table 1 gives an overview of the aliasing
period of the main diurnal and semi-diurnal tidal frequencies on the SWOT Cal/Val orbit, it is completed
by the Rayleigh criterion which provides information on the separability conditions of these waves.
SWOT has been maintained on its Cal/Val orbit for about 6 months, providing slightly more than 4
months of usable data from March to early July 2023, and thus opening up new perspectives for the
study of high-frequency processes and internal tides. What will we learn about internal tides from
SWOT's 1-day orbit? This study provides some answers to this question. It explores and characterizes
the internal tide as seen by SWOT in its unprecedented 1-day orbit and compares it with the high-
resolution empirical tide (HRET) internal tide map from Zaron et al., 2019.



**Table 1**: Period of aliasing (in days, second line) and separability following the Rayleigh criterion (in
days, from the third line to the end) of main tidal waves for SWOT's 1-day orbit.

|  | M2 | S2 | N2 | K2 | O1 | P1 | K1 | Sa | Ssa |
|---|---|---|---|---|---|---|---|---|---|
| Periods | 12.35 | 75.60 | 8.53 | 129.01 | 12.97 | 106.94 | 258.03 | 365.26 | 182.62 |
| M2 | -------- | 14.77 | 27.55 | 13.66 | 258.03 | 13.97 | 12.97 | 12.79 | 13.25 |
| S2 | -------- | ---------- | 9.61 | 182.62 | 15.66 | 258.03 | 106.94 | 95.34 | 129.01 |
| N2 | ------- | ------- | -------- | 9.13 | 24.9 | 9.27 | 8.82 | 8.73 | 8.95 |
| K2 | -------- | ------- | -------- | -------- | 14.42 | 624.89 | 258.03 | 199.47 | 439.51 |
| O1 | ------- | -------- | -------- | ------- | --------- | 14.77 | 13.66 | 13.45 | 13.97 |
| P1 | ------- | -------- | -------- | ------- | -------- | --------- | 182.62 | 151.20 | 258.03 |
| K1 | ------- | -------- | -------- | ------- | ------ | -------- | --------- | 878.92 | 624.89 |
| Sa | -------- | -------- | -------- | ------ | -------- | ------- | ----- | ---------- | 365.22 |
| Ssa | ---- | ---- | ------ | ----- | ------ | ---- | ---- | ---- | ---------- |


This first study based on SWOT data is limited to the Cal/Val track 20 off the Amazon shelf in the
western tropical Atlantic between 2°S and 15°N (Figure 1). The track has been chosen because the
Amazon shelf is one of the hot spots for internal tide generation in the ocean (Arbic et al., 2012; Solano
et al. 2023; Niwa and Hibiya et al.,2011). The region is marked by strong seasonal cycles of
stratification, circulation and eddies that regulate the generation and propagation of internal tides
(Barbot 2021, Tchilibou et al., 2022).  The stratification is modulated by freshwater inflows from
precipitation (under the inter-tropical convergence zone) and rivers (Amazon and Para rivers). The
strong western boundary current, the North Brazil Current (NBC), controls the extension of the
Amazon's plume and develops a double retroflexion into the Equatorial UnderCurrent (EUC, around
2°S-2°N) and the North Equatorial CounterCurrent (NECC, around 5°N-8°N). The barotropic and
baroclinic instabilities of these currents generate some of the eddies present in the region (Aguedjou
et al., 2019). Internal tides generated between the isobaths 100 and 2000 m along the shelf break
propagate mainly from the six sites indicated in Figure 1 (Tchilibou et al.,2022; Assene et al.,2024).
Between March and July, the pycnocline is shallow, the mesoscale activity and currents are low,
consequently, internal tides tend to keep more coherent (Tchilibou et al.,2022). During the rest of the
year, the pycnocline is deeper, mesoscale and current are strong, and, consequently, the incoherence
of internal tides increases as their reflection and advection by the circulation intensifies. As they
evolve, internal tides disintegrate into nonlinear internal solitary waves (Jackson et al., 2012; Alford et
al., 2015). Packets of nonlinear internal solitary waves (ISWs) have been reported along the Amazon
continental shelf and offshore (Lentini et al., 2016; Bai et al., 2021, Brandt et al., 2002; Magalhães et
al.,2016). They are highly active in the area (4-8°N /40-45°W, see Figure 2 of de Macedo et al., 2023)
of concentration of internal tide rays emanating from sites A and D, and they have a seasonal cycle of
occurrence and wavelengths in agreement with those of internal waves (de Macedo et al., 2023).

The orientation of SWOT track 20 in this part of the ocean is such that it intersects three areas with
potentially specific dynamics (Tchilibou et al.,2022). Between 2.5°S and 2.5°N (area 1, Figure 1), the
track is in the path of internal tides generated at points B, C and, to a lesser extent, A.  In area 2,
between 2.5°N and 8°N (Figure 1), the track crosses the zone of interaction between internal tide and
mesoscale. Finally, area 3, north of 10°N (Figure 1), lies on the mid-Atlantic Ridge, where some IT can
likely be generated also.  We'll keep all this in mind when interpreting our results.
In the following, the article is organized as follows: in section 1, we present the data and discuss the
variability of the SLA (Sea Level Anomaly) observed by SWOT along track 20. Section 2 is dedicated to
the comparison between the internal tide signal as seen by SWOT and the HRET model.  An attempt



to separate the coherent and incoherent internal tides is presented in section 3. Then we conclude
with a discussion and perspectives of our results

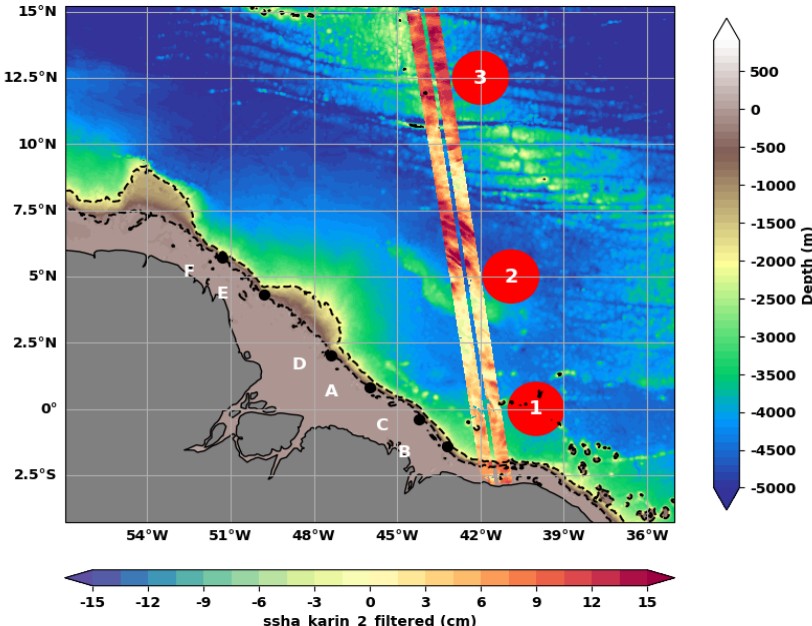


**Figure 1**: Bathymetry (m) off the Amazon shelf in the eastern tropical Atlantic. SLA KaRin (cm) on April
08, 2023, along track 20 of SWOT's 1-day cycle. The main internal tide generation sites are marked by
the letters A to F. The 200 and 2000 m isobaths are dotted. The circles locate area 1 (2.5°S to 2.5°N),
area 2 (2.5°N to 8°N) and area 3 (north of 10°N) along the track.

**1- Data and Variability: Evidence of IT propagation at different scales**

**1.1- Description of the database:**

We use version V0.3 of the L3 SWOT products, released in December 2023. The data, made up of
several variables, are provided on regular horizontal grids of 2 km by 2 km in netcdf or zcoll formats.
Using the variables available in zcoll, we have defined the SLA by equation 1 below:

133        SLA = ssha_karin_2_filtered+ internal_tide_hret - duacs_ssha_karin_2_oi                    (1)

The first term on the right, 'ssha_karin_2_filtered', is the SWOT observation at the two KaRIn swaths
only. We exclude SWOT nadir observation, to focus on the SWOT's potential to observe directly 2D
maps of the ocean.  The ssha_karin_2_filtered has been denoised using data-driven machine-learning
noise reduction and corrected from all the classic physical, instrumental and environmental
corrections applied in altimetry (Dibarboure et al. 2024). The tidal corrections applied are FES2022
model (Lyard et al., personal communication; Lyard et al., 2021) for the barotropic tide and HRET for
the internal tide (Zaron, 2019).  We reintroduced HRET's internal tide SSH (internal_tide_hret), so that
our final SLA consists of the total internal tide signal. The last term 'duacs_ssha_karin_2_oi'
corresponds to the DUACS Maps of Sea Level Anomaly (MSLA) interpolated on SWOT swaths
(Ballarotta et al. 2023; Ubelmann et al. 2015, 2021). It removes the large-scale ocean signals and



particularly the mesoscale eddies that can mask internal waves at these latitudes. On track 20, we have
the SLA from March 29 to July 10, 2023, i.e. 104 cycles with completely or partially filled swaths.
**1.2- Evidence of IT propagation at different scales:**
The snapshots in Figure 2a show very fine-scale crest-like structures superimposed on positive and
negative SLA spaced tens and hundreds of kilometers apart. The scenario repeats itself on the other
cycles (see movie in the supplementary material), indicating that SWOT likely sees internal waves of
different spatial scales.

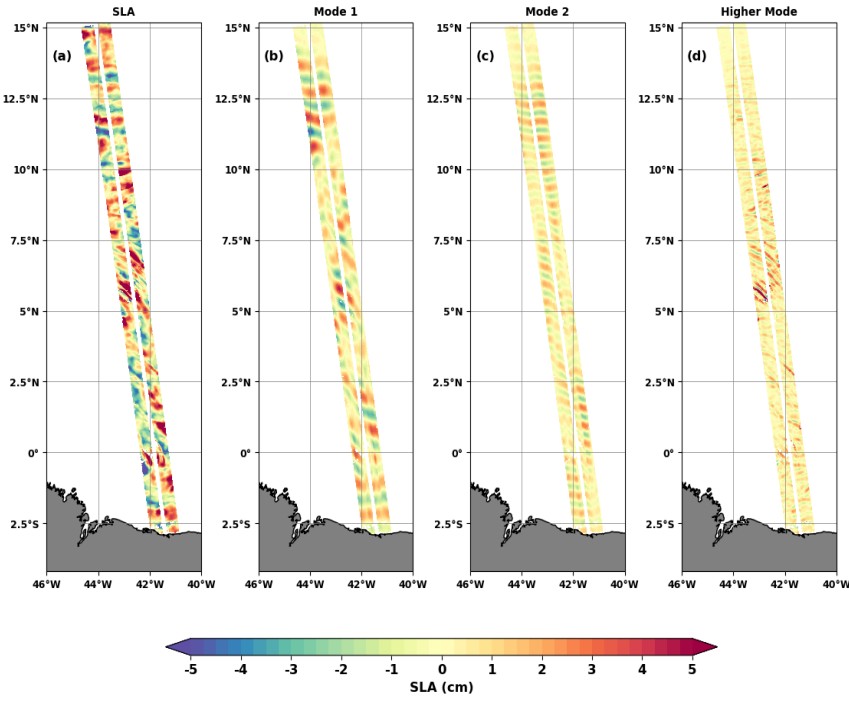


**Figure 2:** Snapshot of SWOT SLA on April 8, 2023. a) Total SLA, b) Mode 1 FFT-filtered SLA (180-90 km),
c) Mode 2 FFT-filtered SLA (80-60 km) and d) Higher mode FFT-filtered SLA (50-2 km).

The wavenumber-frequency (Figure 3a), the wavenumber (Figure 3c) and the frequency (Figure 3d)
spectra of SWOT SLA indicate that the dominant signal is M2 aliased to 12.22 days (see Table 1).  At
the M2 aliased frequency, the energy is greatest between 180-90 km and between 80-60 km (Figure
3a), leading to the spectral peaks in Figure 3c.  These two wavelength bands correspond well to the
theoretical baroclinic mode 1 and 2 scales expected for the internal tide in this region (Zhao, 2021).
We isolated the SLA for these two wavelength bands using FFT filtering along the track (approximately
latitudinal direction). Snapshots of the Mode 1 and Mode 2 SLA are shown in Figures 2b and 2c for the
same day as Figure 2a, revealing more of the SLA's wave-like behavior.
Figure 2d shows the FFT-filtered SLA between 50-2km. This band contains all the small-scale
structures, including the very remarkable and intense one that appears as wave crests on the SLA. On
the wavenumber-frequency spectrum (Figure 3a), the energy maximum at frequency M2 extends to



scales smaller than 50 km. According to Barbot et al., (2021), this could be associated to internal tide
of mode 3, mode 4 and mode 5. We therefore consider the 50-2 km band as consisting of higher
modes.

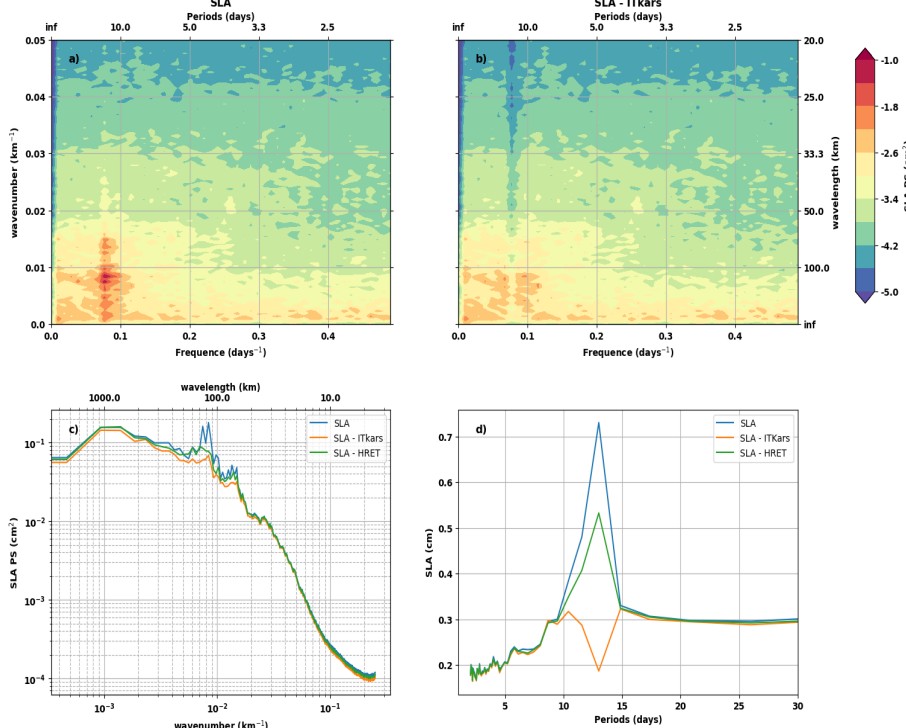


**Figure 3:** Wavenumber-frequency spectra of the total SLA (a) and ITKars detided SLA (b). Wavenumber
(c) and frequency (d) spectra of the total SLA (in blue), ITKars detided SLA (in orange) and HRET detided
SLA (in green). ITKars is the internal tide model derived from SWOT KaRIn data (cf in section 2.1)
**1.3- Variability analysis of IT observations:**

Analyses of SLA variability are completed by calculating the standard deviations of the total and the
spatially FFT-filtered SLAs in the wavelength bands defined above. Over the Cal/Val period, SLA varies
between 1 and 5 cm under track 20 (Figure 4a). Apart from the area very close to the coastline, there
are three main patches of maximum variability, each located in one of the dynamic areas highlighted
in the introduction. The maximum variability of the SLA in area 1 (2.5°S-2.5°N) is mainly due to the
regular mode 1 internal tide flux likely coming from sites A, B and C (Figure 4b). Mode 2 and higher
modes contributions are secondary (Figures 4c and 4d) in area 1. Higher modes have a major impact
on the variability in area 2 where they make the SLA vary by 2 to 3 cm (Figure 4d), i.e. almost of the
same order as mode 1 in the same area. As area 2 is far from the Amazon shelf, the higher modes here
likely originate from interference between mode 1 and mode 2 semi diurnal IT (Solano et al.,2023). In
area 3, SLA variability is driven mainly by mode 1 and mode 2.





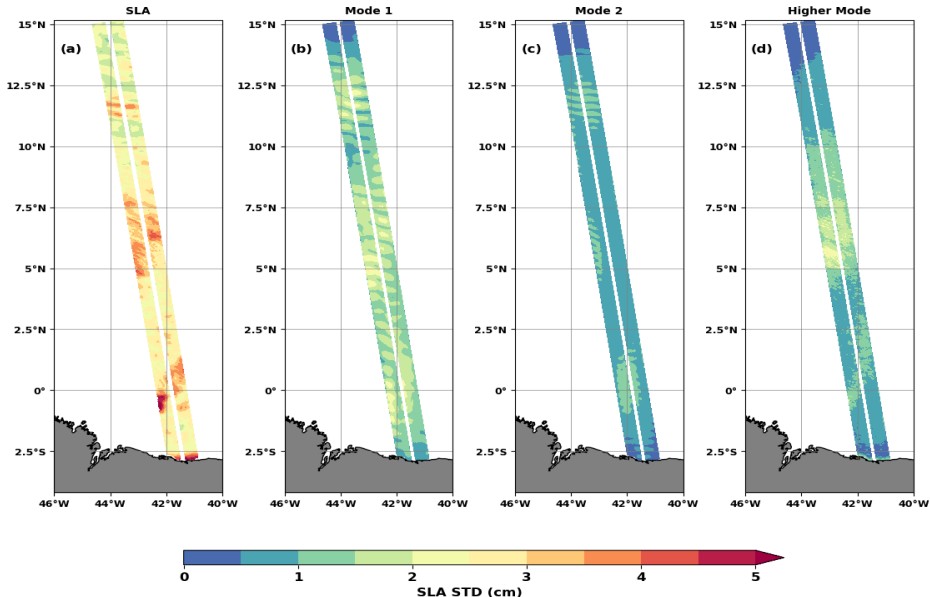

**Figure 4:** Standard deviation (in cm) of the total (a) SLA, and mode 1(b), mode 2 (c) and higher mode (d) FFT-filtered SLA.

**2- Comparison between SWOT and HRET: coherence and predictability of internal tides**

In this section, we evaluate the coherent internal tide from SWOT KaRin Cal/Val data for the main semi-diurnal frequencies, compare the M2 results to the HRET model and calculate an internal tide incoherence coefficient.

**2.1- The M2, S2 and N2 coherent internal tides from SWOT: ITkars model**

In Table 1, 5 waves (M2, N2, S2, O1 and P1) have aliasing periods shorter than the 104 days corresponding to the total length of our SWOT SLA series, and are a priori of interest for our analysis. But given the Rayleigh criterion between them in Table 1, it is reasonable to restrict ourselves to the three semi-diurnal waves. Using harmonic analyses, the coherent internal tide is extracted at each swath point that has at least 80 valid cycles over the entire SWOT Cal/Val observation period. In the following, ITkars (IT from KARin Swot) refer to SWOT estimation of IT.





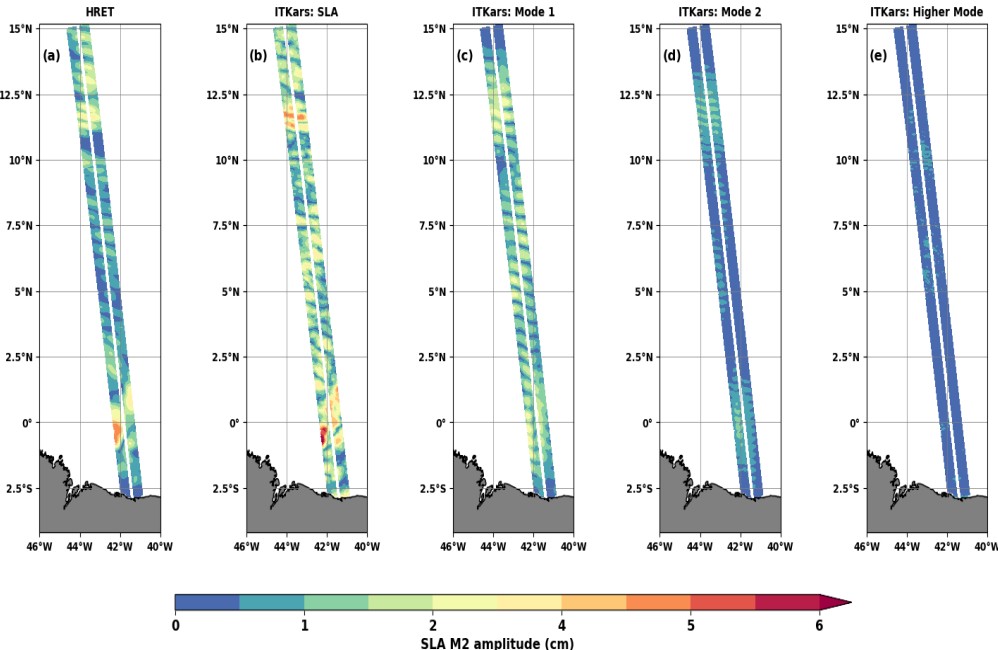

**Figure 5:** The amplitude (in cm) of the M2 internal tide from the HRET model (a) and the ITkars (b to e) model over the cal/val period. ITkars is derived by harmonic analysis of the total SWOT SLA (b) and FFT-filtered SWOT SLA for mode 1 (c), mode 2 (d) and higher mode (e) SLA. Only swath points with at least 80 valid cycles were analyzed.

The amplitudes of the coherent internal tide at M2 frequency are presented in Figure 5 for both HRET and ITkars models. We first performed the harmonic analysis of the total SLA (Figure 5b) and repeated the harmonic analyses for each of the FFT-filtered SLAs (Figure 5c to e). The HRET model (Figure 5a) and the ITkars model based on the total SLA (Figure 5b) are similar in terms of spatial distribution, although HRET has smoother and lower amplitudes. In areas 1 and 3, ITkars shows spatial features identical to those already observed on the standard deviation in Figure 4a. So, the maximum variability for these two parts of the SWOT track is indeed due to the M2 coherent internal tide. The discrepancies between standard deviations (Figure 4) and internal tide amplitudes (Figure 5) are best seen by directly comparing the maps for the different modes or wavelength bands. In area 2, the amplitude of the coherent internal tide is less than 1.5 cm for the higher modes (Figure 5e), whereas at these scales the standard deviation is maximal (Figure 4d). The high variability of the SLA found in area 2 is evidently related to internal tide incoherency.

S2 and N2 are not available in HRET products, so we show only ITkars results in Figure 6. Both waves have smaller amplitudes than M2 and do not have the same structure as the latter. As the semi-diurnal S2 and N2 IT should have similar patterns to M2, those results indicate that these frequencies are certainly contaminated by other tidal waves due to bad separability on the available period (see Table 1) and mesoscale also.





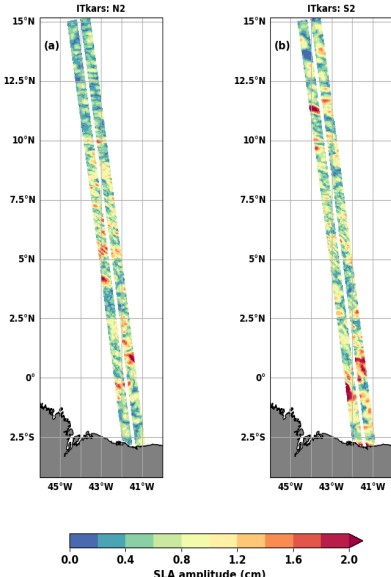

222

**Figure 6**: The amplitude (in cm) of the N2 (a) and S2 (b) internal tide of the ITkars model derived by harmonic analysis of the total SLA over the Cal/Val period. Only swath points with at least 80 valid cycles were analyzed.

**2.2- Predictability: detiding, incoherency and variance reduction analysis**

To go a step further in the comparison between HRET and ITkars, we used the tidal estimation of ITkars (described in previous subsection) over the entire Cal/Val period to detide the total SLA observed by SWOT. To stay in line with HRET and considering the results in Figure 6, the ITkars detiding is limited to M2. The 2D wavenumber-frequency spectrum of the detided SLA is shown in Figure 3b, and the associated wavenumber and frequency spectra are shown as orange lines in Figures 3c and d. In these figures, the green line spectra correspond to the detiding based on HRET model.

When detiding with ITkars, the energy spectrum (Figure 3b) decreases around the aliased frequencies of M2 (around 13 days, see Table 1). The mean of SLA amplitude along the SWOT swaths drops by ~0.5cm (71% of 0.7 cm of the total SLA) after ITkars detiding at M2 frequency. With HRET (Figure 3d) correction, the mean of SLA amplitude is reduced by 28% at the M2 frequency, i.e. about twice less than after ITkars detiding. One can also notice that periods over 15 days and below 5 days are not impacted by the ITKars correction. On the wavenumber spectra, the peaks of modes 1 and 2 are reduced but remain visible whatever the detiding applied (Figure 3c). ITKars reduces them slightly more than HRET, although ITKars also seems to affect some of the larger scales of the SLA, probably indicating that the accuracy of the tidal estimates is limited by the short SWOT Cal/Val time series available.

We have integrated the wavenumber spectra over all wavelengths, between 180 and 60 km for mode 1, between 80 and 60 km for mode 2, between 50 and 2 km for the higher modes, and finally over wavelengths greater than 180 km for the large scale. The derived standard deviations are presented in Table 2 for the total SLA and the SLA detided with ITkars or HRET, as well as the percentages expressing the rate of variance of the detided SLA compared to the total SLA. The higher is the standard deviation of the detided SLA or the percentage in Table 2, the less efficient is the detiding. According to Table 2,





the application of the M2 internal tide prediction of each of the models removes very little variance
from the SLA, nevertheless ITkars is more efficient than HRET especially at mode 1 and mode 2 scales.
For these scales 76% and 84% of the SLA is likely to be incoherent internal tide after correction by
ITkars. For the higher modes, Table 2 agrees with Figures 5 and 6: the M2 correction has no effect at
these scales. ITkars has a greater impact on large SLAs than HRET. The reason for this is not clear to us.

The better performance of ITKars is not surprising, since the detiding is performed over the same
period as the harmonic analysis. Another way to compare ITkars and HRET predictions is to calculate
the standard deviation (STD) reduction (see equation 2 below) first over an analysis period and second
over a validation period. To this purpose, we split the SWOT database in two: the period 1, consisting
of the first 70 cycles, and period 2, consisting of the last 34 cycles. We repeated the M2 harmonic
analysis on period 1 and derive the "ITkars_p1" model (p1 indicating period 1). The internal tide model
is not derived from period 2, the period 2 data is independent from period 1. Period 2 can be taken as
a validation period.

$$\text{STD reduction} = \text{std (SLA} - \text{ITkars\_p1)} - \text{std (SLA} - \text{HRET)} \qquad (2)$$

The SLA has been corrected with M2 from ITkars_p1 on the one hand and M2 from HRET on the
other, over periods 1 (Figure 7a) and 2 (Figure 7b); the STD reduction is determined as in equation 2.
A negative std reduction indicates that detiding with ITKars_p1 reduces more variance than HRET, it is
mostly the case in Figure 7a for period 1. Positive values dominate in period 2, indicating that ITkars_p1
predictions fail to produce a realistic internal tide pattern over the independent period. We notice
that the increase in SLA variance by ITkars_p1 during period 2, is stronger in area 2, where the higher
modes greatly contribute to SLA variability (see Figure 4). Once again, these results can likely be
explained by the strong incoherency of the internal tide in this area, but also by the short time-series
used for the tide estimation which induces some uncertainty in the along-track tidal model due to
some remaining separation problems and residual small scale ocean contamination. The spectra, the
Table 2 and the STD reduction analysis are unanimous on the high degree of internal tide incoherency
under track 20 off the Amazon shelf, particularly for the very small scales and very high frequencies.
Can we hope to separate the coherent and incoherent components of the internal tide under this
SWOT track, and then improve our estimate of the coherent internal tide?

**Table 2:** Comparative table of the standard deviations of total SLA and SLA detided with HRET or ITkars.
Standard deviations are obtained by integrating the spectra of Figure 3c on different wavelength bands
(in cm). The ratio between detided SLA and total SLA, computed as a percentage, is given in
parentheses.

|  | All wavelengths | Large scales >180km | Mode 1 180 - 90km | Mode 2 80 - 60km | Higher modes 50 - 2km |
|---|---|---|---|---|---|
| Total SLA | 1.82 | 1.07 | 1.03 | 0.58 | 0.74 |
| Detided ITkars | 1.6 (88%) | 0.99 (93%) | 0.78 (76%) | 0.49 (84%) | 0.71 (96%) |
| Detided Hret | 1.71 (94%) | 1.04 (97%) | 0.89 (86%) | 0.54 (93%) | 0.73 (99%) |



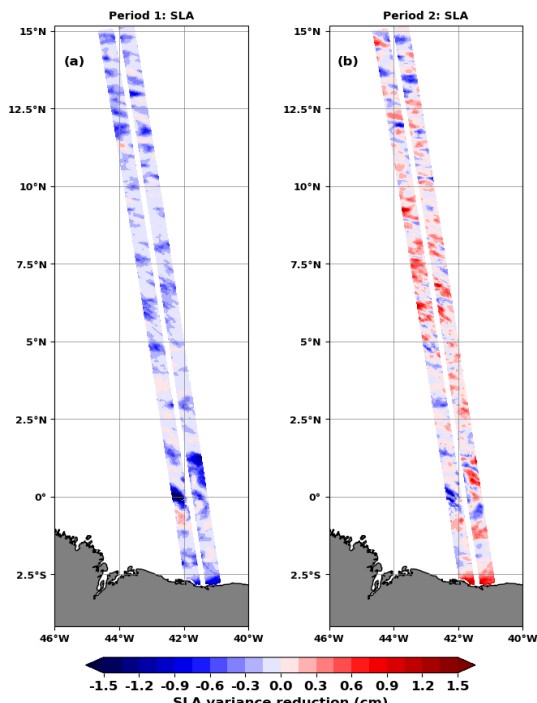

**Figure 7:** STD reduction (in cm) for SWOT SLA when using M2 ITkars_p1 internal tide correction or M2 HRET correction. The std reductions are calculated over period 1 (left) from late of March to early June (first 70 cycles) and over period 2 (right) from early June to early July (last 34 cycles).

**3- An attempt to improve the estimation of coherent internal tide from SWOT Cal/Val data: Using principal component analysis (PCA) to separate SLA content**

**3.1- Separation using PCA**

PCA, also known as EOF (Empirical Orthogonal Function), is a statistical analysis technique for reducing the dimensionality of a data set (Jolliffe, 1986). Applied to geophysical data, PCA separates the total signal into independent spatial patterns associated with independent temporal components (Principal Component) and gives a measure of the relative importance of each pattern (a percentage of the total variance). The first principal components (PC) capture most of the variance in the data and generally have a repetitive and persistent structure, they behave approximately like the stationary component of the signal. On this basis, we believe that PCA applied to our total SLA can help better isolate the coherent internal tide (which is stationary) from the remaining residual tidal and non-tidal signals observed by SWOT. We performed the PCA on all 104 cycles of the SWOT KaRin total SLA. At each point in the swath, we filled in the missing value with the local time mean, then normalized the time series to ensure that the time mean, and the standard deviation became zero and one respectively. The covariance matrix is calculated on the normalized SLA, the PCA focuses on eigenvalues and not absolute values.

The two leading PCA modes shown in Figure 8 account for 12.3% (PCA1, Figure 8a and c) and 9.1% (PCA2, Figure 8b and d) of the total variance. Their spatial patterns correspond to IT structures: on





PCA1 (Figure 8a) the IT is intensified in area 1 and area 2, while PCA2 (Figure 8c) is characterized by an
increase of the IT intensity in area 2. PCs have 12–13 days oscillations, with amplitude modulations
around 70 days (Figure 8b and 8d), therefore recalling the aliasing periods of M2 and S2 waves (see
table 1). To get a more precise idea of the wavelengths and frequencies contained in PCA1 and PCA2,
we reconstructed the SLA for both components (SLA_pca1 and SLA_pca2) and calculated the spectra
shown in Figure 9 (blue line for PCA1 and orange line for PCA2).

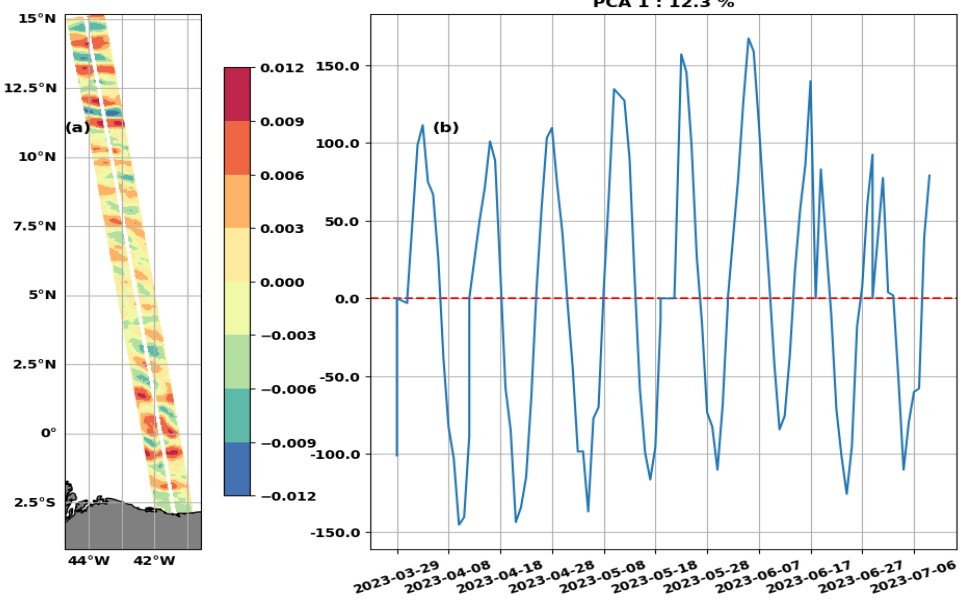


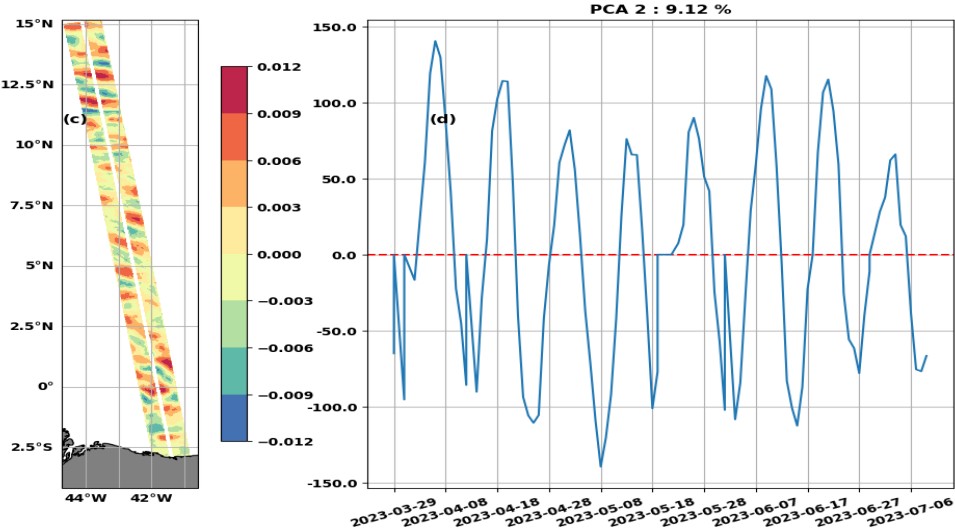


**Figure 8:** Spatial (left) and principal (right) components of PCA1 (top) and PCA2 (bottom) of the SLA
along SWOT swaths over the Cal/Val period.



The wavenumber spectra (Figure 9a) indicate that PCA1 and PCA2 consist mainly of mode 1 (180-90
km) and mode 2 (80-60 km) IT.  A peak that could be associated with mode 3 stands out on the PCA2
spectrum, but overall, the energy levels of both spectra remain low for higher modes (50-2 km). The
frequency spectra (Figure 9b) confirm that M2 is the dominant signal. At this frequency the mean SLA
amplitudes are 0.52 cm for PCA1 (71% of 0.73 cm of the total SLA reported in solid black line in Figure
9b) and 0.45 cm for PCA2 (61%).  Amplitudes are low for other frequencies, and 104 cycles are not
enough to observe 70-days modulation on the frequency spectra. Given the wavenumber and
frequency spectra, we can say that PCA1 and PCA2 are two complementary representations of the
propagation and evolution of the M2 dominant internal tide, so they can be merged to form a single
signal.  We have summed SLA_pca1 and SLA_pca2 into SLA_pca_L2 (L2 refers to lower or equal to 2).
A snapshot of SLA_pca_L2 is shown in Figure 10a for the same cycle as in Figure 2. Interestingly, the
SLA reconstructed with PCA1 and PCA2 have similar patterns to the mode 1 and mode 2 FFT-filtered
SLAs (Figures 2b and 2c).
Between PCA3 and PCA12 the variance explained is less than 3.5% per PCA, from PCA13 onwards,
the variance becomes less than 2% (not shown). The PCs are a mixture of several wave frequencies,
with M2 of lower intensity than in PCA1 and PCA2, high frequency (faster than 10 days) and low
frequency (15, 17 or even 25 days). It is difficult to associate the spatial patterns of these PCAs with
the propagation of a persistent IT in time and along the track, or even with a mode of ocean variability
to our knowledge; some patterns also resemble residual noise from the processing of raw SWOT data.
We grouped PCA3 to PCA104 into SLA_pca_G2 (G2 for greater than 2). The small-scale structures
detected in Figure 2a are clearly visible on the snapshot of the SLA_pca_G2 in Figure 10b. Figures 10a
and 10b are complementary, as the PCA acted as a filter. The total SLA is now split into SLA_pca_L2
and SLA_pca_G2. The spectra of the total SLA corrected with M2 from the ITkars in section 2 are
reproduced as black dotted lines in Figure 9; at all frequencies and wavelengths, they overlap well with
the spectra of SLA_pca_G2 (green solid lines in Figure 9). Therefore, SLA_pca_L2 is more suitable for
building a model of coherent internal tide.

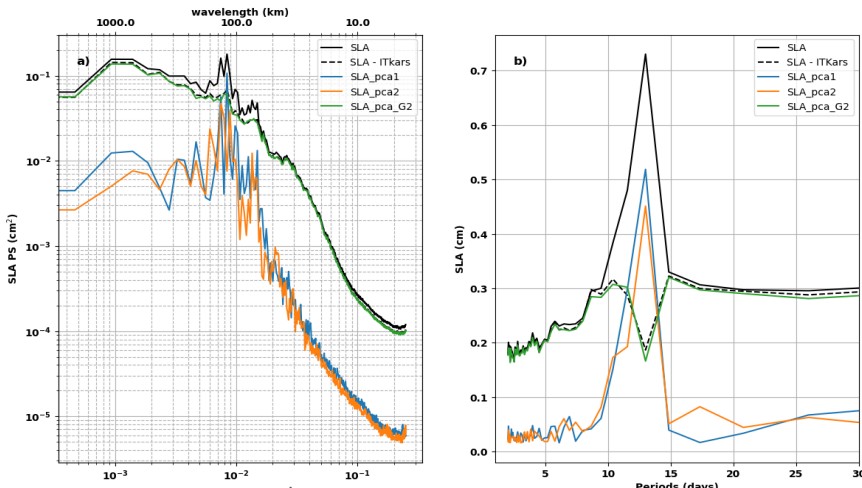


**Figure 9:** Wavenumber (a) and frequency (b) spectra of  SLA_pca1 ( in blue),  SLA_pca2 (in orange)
and SLA_pca_G2 (in green). SLA_pca_G2 is the sum of the SLAs of PCAs greater than 2. The spectra of
total SLA (black solid line) and SLA - ITkars (black dotted line) from figure 3 are reported here.



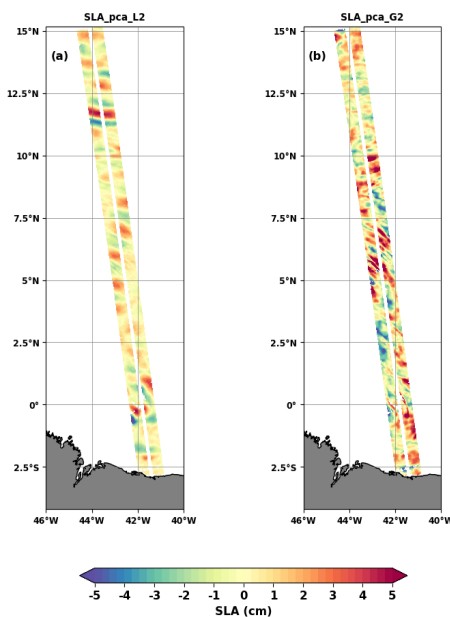


**Figure 10:** Snapshot of SWOT SLA_pca_L2 (a) and SLA_pca_G2 (b) on April 8, 2023 (as in Figure 2). SLA_pca_L2 is the sum of the SLAs of PCs less than or equal to 2 (PC1 and PC2).


**3.2- ITkars_pca internal tide model**

We have performed the harmonic analysis of SLA_pca_L2 at the semi-diurnal frequencies M2, N2 and S2 (Figure 11). The resulting internal tide model is referred to as ITkars_pca to distinguish it from ITkars based solely on harmonic analysis of SWOT Karin data. Compared to Figure 6 corresponding to ITkars, the ITkars_pca internal tide maps for N2 (Figure 11a) and S2 (Figure 11b) are cleared of small scales, and the patterns for both waves are now close to that of M2 as expected (Figure 11c and 5b). At first glance, there seems to be no difference between ITkars (Figure 5b) and ITkars_pca (Figure 11c) for M2, but by making the complex difference between the two signals we deduce the amplitude shown in Figure 11d, which is equivalent to the amplitude of the harmonic analysis of SLA_pca_G2 at M2. As with N2 and S2, Figure 11d shows that ITkars also contains an additional signal dominated by small scales, and which does not resemble the classic internal tide.

361



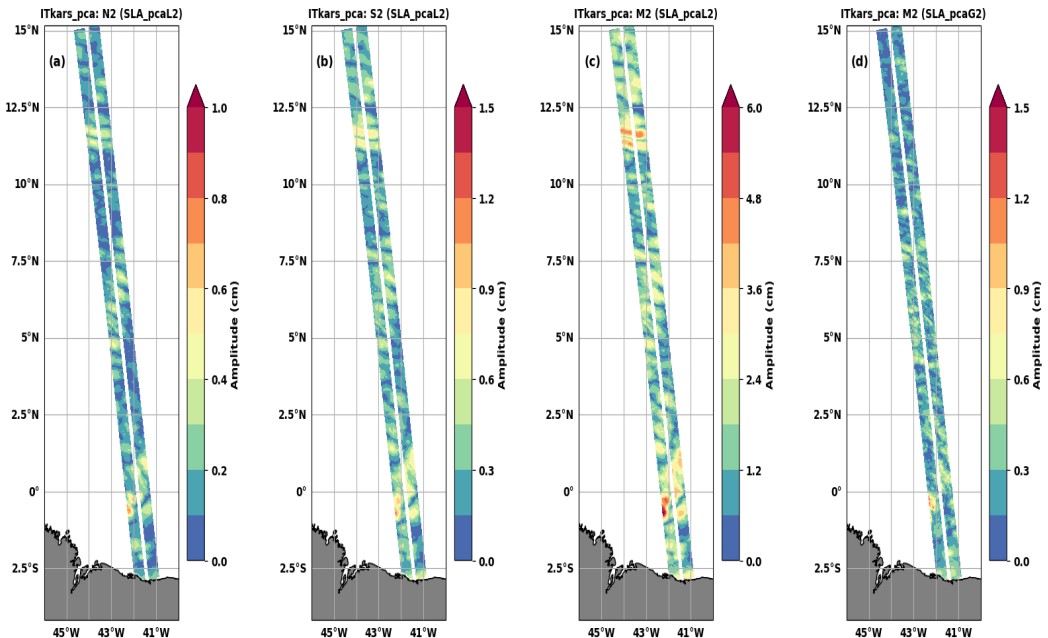

Figure 11: The amplitude (in cm) of the internal tides N2 (a), S2 (b) and M2 (c and d) of the ITkars_pca model derived by harmonic analysis of SLA_pca_L2 (a to c) and SLA_pca_G2 (d) over the cal/val period. SLA_pca_L2 is the SLA based on PCA1 and PCA2, SLA_pca_G2 is compiled from PCA3 to PCA104. Only swath points with at least 80 valid cycles were analyzed.

The origin of the extra signal contaminating ITkars could be dynamic or numerical. Dynamically, these could be very intense non-linear waves, soliton, or incoherent internal tide, which are retained in the harmonic analysis of section 2 due to the short length of the time series. On the numerical side, noise linked to the pre-processing of SWOT data cannot be ruled out. Another source of contamination could also be the DUACS correction we apply beforehand to distinguish internal tide.

Finally, the capacity of ITkars_pca to detide the SLA is tested. As in section 2, M2 ITkars_pca is estimated over period 1 (ITkars_pca_p1) and then validated over period 2. ITkars_pca_p1 is used to detide both the total SLA and the SLA_pca_L2 from which it is built. The variance reduction (standard deviation, see equation 2) are shown in Figure 12, while Table 3 summarizes the statistics for both periods. On the total SLA, ITkars_p1 (Figure 7a), and ITkars_pca_p1 (Figure 12a), have equivalent performance in period 1, with both models correcting for 15% and 14% of SLA variance respectively (Table 3). The transition from ITkars to ITkars_pca is characterized by an additional decrease of the residual variance of the total SLA over period 2 (from 95% to 91%, Table 3).




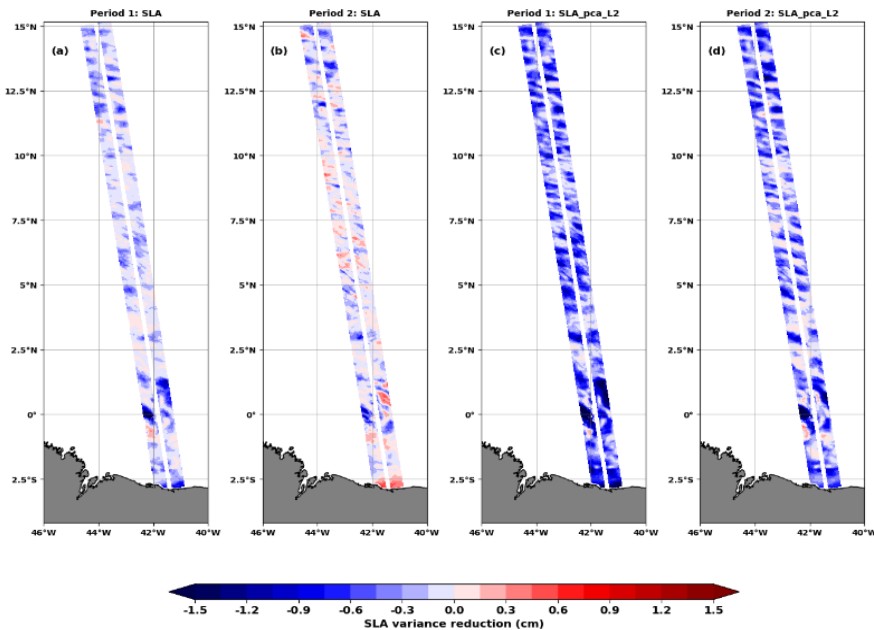

380

**Figure 12:** STD reduction (in cm) for either SWOT SLA (a and b) or SLA_pca_L2 (c and d) when using
M2 ITkars_pca_p1 internal tide correction or M2 HRET correction. The std reductions are calculated
over period 1 (a and c) from late of March to early June (first 70 cycles) and over period 2 (b and d)
from early June to early July (last 34 cycles).

385

There remains 74% (period 1) and 79% (period 2) of the variance of SLA_pca_L2 when detiding with
HRET, which indicates that HRET is not efficient enough even on these SWOT data consisting a priori
of coherent internal tide only. Unsurprisingly, the STD reductions of SLA_pca_L2 are negative when
comparing HRET to ITkar_pca in Figure 12c and 12d. In period 1, the STD of SLA_pca_L2 decreases
from 1.33 cm to 0.53 cm (decrease of 60%) after detiding with ITkars, and to 0.42 cm (decrease of 68%)
if ITkars_pca is applied (Table 3). In period 2, there is a 0.1 cm difference between STDs when
SLA_pca_L2 is detided with either ITkars or ITkars_pca (10% more decrease with ITKars_pca). As
described above, ITkars_p1 is the sum of ITkars_pca_p1 and a residual similar to the one seen in Figure
11d; since this residual signal is absent from SLA_pca_L2 by construction, the STD gap between both
corrections ITkars and ITkars_pca would give an estimation of the level of variance linked to the
residual signal in ITkars. Note that SLA_pca_L2 is only corrected for M2 in this variance test, and even
using ITkars_pca there are still signals from waves that have not been evaluated, such as the 70-days
modulation. Overall, PCA as a preliminary step before harmonic analysis has a positive impact on the
internal tide model and on the quality of detiding of SWOT 1-day SLA observations over the Cal/Val
period.

**Table3:** Comparative table of standard deviations (cm) of SLA and SLA_pca_L2 detided with either M2
HRET, M2 ITkars or M2 ITkars_pca models over period 1 (from late March to early June 2023, the first
70 cycles) and period 2 (from early June to early July 2023, the last 34 cycles). ITkars_p1 and
ITkar_pca_p1 models were built on period 1 and validated on period 2. The ratio between detided SLA
and total SLA is indicated in the parentheses (in percent).




|  | Period 1 | | Period 2 | |
|---|---|---|---|---|
|  | SLA | SLA_pca_L2 | SLA | SLA_pca_L2 |
| no IT correction | 2.56 | 1.33 | 2.79 | 1.15 |
| HRET | 2.39 (93%) | 0.99 (74%) | 2.64 (95%) | 0.91 (79%) |
| ITkars_p1 | 2.18 (85%) | 0.53 (40%) | 2.65 (95%) | 0.63 (55%) |
| ITkars_pca_p1 | 2.21 (86%) | 0.42 (32%) | 2.55 (91%) | 0.52 (45%) |


**4- Discussion and perspectives**
In this study, we explored and characterized the internal tide signal in SWOT KaRIn observations over
the Cal/Val period (1-day orbit) between late March and early July 2023 (104 cycles) and along the
track 20 located off the Amazon shelf in the tropical Atlantic between 2°S and 15°N. The internal tide
as seen by SWOT is a mixture of several spatial scales, including baroclinic modes 1 and 2 defined by
wavelengths between 180-90 km and 80-60 km respectively. SWOT also sees very intense fine-scale
structures (wavelengths between 50-2 km) that we have associated with higher baroclinic modes,
including modes 3, 4 and 5 according to Barbot et al., (2021). As a result, SWOT seems to live up to
expectations, providing a direct 2D view of the internal tide sea surface signatures and even access to
smaller scales.
Our approach to extract the internal tide signal through the 1-day SWOT data consisted firstly of
filtering the large scale (including the mesoscale) by subtracting the DUACS MSLA from the SWOT
observations; then we reintroduced the internal tide correction HRET from Zaron (2019) to obtain an
SLA consisting of the total internal tide signal and finally. We either performed the harmonic analysis
(as in section 2) or proceeded upstream to the PCA before the harmonic analysis (as in section 3).  The
internal tide model based on harmonic analysis of SWOT KaRin data was referenced ITkars (Internal
Tide from KaRin Swot), the one obtained by combining PCA and harmonic analysis ITkars_pca. We
focused on the semi-diurnal frequencies M2, S2 and N2.
The ITkars and ITkars_pca models were found to be close to the M2 HRET model based on nearly 25
years of conventional altimeter (nadir) observations. The similarities between models based on SWOT
Karin and model with conventional altimeter are partly linked to the fact that SWOT data are analyzed
over March to July during which the internal tide is most stable and coherent off the Amazon shelf
(Tchilibou et al., 2022).  One consequence of analyzing SWOT data over this short 104-day window is
that the amplitude of the internal tide is stronger with SWOT estimation than with HRET. This result is
logical since the intensity of the coherent internal tide depends on the length of the time series
analyzed: a longer time series allows a better estimate of the coherent signal which is therefore
smoother (Ansong et al.,2015; Zhou et al.,2015; Nash et al.,2012). The separation of M2 from O1 is not
ensured with 104 cycles of SWOT 1-day data, however, in this region the amplitude of the internal tide
is negligible at O1 compared to M2 (see Figure 1 in Tchilibou et al.,2022), so M2 ITkars_pca is thus
quite reliable.
The maps of N2 and S2 highlighted the contamination of ITkars by signals other than the coherent
internal tide, and particularly by very small scales. We hypothesize that the contamination is due to
the leakage of nonlinear waves, part of incoherent internal tides, and ocean variability in the harmonic
analyses. Regarding ocean variability, a part is not captured by DUACS and therefore was not subtract
from the SLA, moreover the prior subtraction of the mesoscale as we did is in itself a source of error in
the estimation of the internal tide (Zaron and Ray, 2018). One way to reduce the effects of
contamination by ocean circulation would be to apply a simultaneous internal tide and mesoscale
inversion method as proposed by Ubelmann et al. (2022). The combination of PCA and harmonic



analysis gives semi-diurnal ITkars_pca maps (M2, S2 and N2) with similar patterns. The amplitude of N2 ITkars_pca deduced from SWOT is of the same order as that in the new product HRET14 (E. Zaron personal communication). The result is encouraging for S2, especially as the length of the 1-day observations is not sufficient to correctly separate it from waves such as Sa and Ssa, whose periods are identical to those of the annual and semi-annual variation of the ocean. A longer time series is needed to better separate the internal tide components from SWOT observation, and we will consider analyses of the 21-day SWOT science orbit data when the time series will be long enough.

PCA has improved our estimate of the internal tide model from the SWOT KaRin data. From the PCA we kept the first two main modes (PCA1 and PCA2) and considered them as the coherent internal tide given their fairly stationary character. Thus, the coherent internal tide accounts for 21.42% (12.3 of PCA1 and 9.12 of PCA2) of SLA variance in 1-day SWOT observations, a proportion in line with the studies of Zaron (2017) and Egbert and Erofeeva, (2022) in this region. The coherent internal tide isolated through the PCA consists of mode1, mode 2 noticeable in PCA1 and PCA2, and mode 3 noticeable in PCA2. The fact that the coherent internal tide signal is projected onto two main modes of the PCA is an open question. The principal components of PCA1 and PCA2 are shifted by 3 to 4 days, about a quarter of the aliased frequency of M2, which could correspond to a phase quadrature, as there is between the imaginary and real parts needed to reconstruct a sinusoidal signal. Another possibility is that PCA1 and PCA2 represent the same phenomenon, with the peculiarities of area 2 in the middle of the swaths, when the internal tide is moderate for PCA1 and when it intensifies for PCA2. This type of PCA behavior is observed in the case of ENSO studies in the Pacific (Takahashi et al., 2011). The peaks on the wave number spectrum of PCA1 and PCA2 are shifted by few kilometers at the mode 1 and mode 2 scales, suggesting a change in wavelengths relating to changes in stratification conditions as suggested by Barbot et al. (2021). A longer series of Cal/Val observations could have helped to better distinguish PCA1 from PCA2.

The principal components of PCA1 and PCA2 also give an overview of the daily variability of the internal tide amplitude, a result that is currently unattainable with conventional altimetry missions. The opportunity to learn more about the temporal variability of the internal tide using a single high-resolution mission is lost, or at least postponed, with SWOT's switch to its 21-days scientific orbit. One of the limitations of using PCA to analyze SWOT data is probably its sensitivity to track length. The total variance is distributed differently in the principal components depending on whether the track is long or short, or whether ocean dynamics change significantly along the track. It would be interesting to look at this point in the perspective of a global model, for example. We are curious to know how the PCA will behave in the case of multi-track use, and at their crossing points.

In the context of 1-day SWOT observations, the use of PCA can be useful in determining wave frequencies of interest for the development of the coherent internal tide model. The combination of PCA and harmonic analysis further reveals the observational potential of SWOT. We are currently working on other SWOT tracks in various ocean regions to test the robustness of our method combining PCA and harmonic analysis. We also plan to explore in situ observations of the SWOT Cal/Val and other databases to understand better our results. Work remains to be done to confirm the presence of mode 3 in the coherent internal tide signal in this region. The incoherence of the internal tide and its interaction with the circulation are other issues to be addressed with these SWOT data.



**Authors contributions:** This work is part of the Marée - SWOT project funded by the CNES at CLS. MT's
work and analyses are supervised by LC and FL. Conceptualization: ML, LC, FL, CU. MT wrote the
paper with contributions from all co-authors.
**Competing interests:** The contact author has declared that none of the authors has any
competing interests

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
