# Peer review of "SWOT Cal/Val Mission Data"

_EGUsphere, 2024_

## Author Response (AR1)

The paper is useful analysis of internal tides seen in new SWOT data along a pass off the Amazon Shelf. It is a good paper and shows that exciting new tide results can come from flying an altimeter that measures a swath, rather than a single nadir line. The authors develop a PCA methodology that improves standard harmonic analysis for tides, showing reasonable results for S2 and N2 that complement M2. This is a useful advance that others may also find helpful to their work, with SWOT or otherwise.

My only criticism, which is minor, is that the sections of the paper that split up the SWOT 104 cycles into 70/34 segments are somewhat complicated and not easy to follow. Furthermore, I didn't see the point, except it emphasized how much improvement is obtained with the PCA method, which we already saw anyway. It does show that variance can be removed from times not used in the analysis, but it seems doubtful this will apply in general, since other times of the year have different levels of coherence/incoherence, as authors stressed. The paper could be shorter, and easier to follow without this 70/34 split. However, I'm sure the authors have an argument for keeping it.

Even if the 70/34 split material is kept, I think the paragraph starting at line 386 is very difficult to follow and could be rewritten for clarity.

>>> Thank you for reading our article and recommending it for publication. We are aware of the difficulty in understanding the part where the data is broken down into 70/34. This is an essential step in our work to validate our internal flood estimates. To address your concerns, we have reorganised the paper. Section 2.2 and the part of the paper starting from line 372 have been merged and a new section (Section 4) has been created. Section 4 has been written at the very end, before the conclusion, and isolated from the parts of the paper that deal with internal tides estimation. We hope that this change will make it easier to understand these analyses.

A few other very minor items to address are as follows:

Figure 8: There are some strange kinks in the PCA time series, such as near May 20. Is this caused by gaps in SWOT data? If so, then the blue lines should have a break in them. If not, I am surprised these kinks didn't corrupt the spectra of Figure 9.

>>> It's true that some data is missing, and this may be the cause of the break in the PCA time series. The break would have been complete if it had been present in all cycles. Missing values can be seen as masked values in the spectral analysis, their effect is less significant as the final spectrum is a mean.

44 - Chelton et al.(1998) had nothing to say about the SSH of internal tides. Suggest removal.

>>> the reference has been removed

209 - "HRET has smoother and lower amplitude" - This is expected since I believe HRET is a mean over many years. This should be acknowledged. (It is acknowledged in the final section, but it should be stated here, too.)

>>> We've added a paragraph at the end of sub-section 1.1 to describe HRET. And we have completed the sentence with : ..."because it represents a mean on many years of altimetry data."

184 - high modes originate from interference? I do not see what evidence authors had to make this statement.

>>> The phrase has been replaced by: As area 2 is far from the generation sites of the Amazonian shelf-break, the higher modes here are likely to originate from desintegration of mode 1 and mode 2.

133 - final term in Eq (1) - The "karin_2_oi" phrase on to duacs_ssha; is this the standard DUACS product, or some kind of special version?

>>> This is the binary format available on CNES cluster (TREX): it contains same data as the netcdf files in addition to some complementary fields (MSLA DUACS …).

421 - "and finally" is a phrase not connected to anything.

>> The word has been removed.

Figure 3: The spectra in panels (b) and (d) suggest that harmonic analysis is actually extracting too much energy. It may be one reason that PCA is able to improve the simple harmonic estimates.

>>> Yes, the spectra suggest that HA extracts too much energy particularly for large scales, and some complementary information is given in Fig 6, with the direct estimations of N2 and S2 frequencies which are very noisy. The PCA allows improving the harmonic estimates because it allows a better separation between the coherent and non-coherent parts of the variability.

490 - author initials ML?

>> Thanks for the remark, it's been corrected (MT)

Responses to RC2 :

The authors analyse approximately 16 weeks of SWOT fast-sampling phase data in a region off the Amazonian shelf, where strong internal tidal (IT) generation and propagation occurs, and caracterize the IT field in this region as retrieved from SWOT. The analysis proposes different methods to extract the internal tidal field from the data, including the use of Proper Orthogonal Decomposition in combination with harmonic analysis, and includes a comparison with the HRET product. The impact of the incoherence of the IT on the reconstruction is also addressed. The results are very interesting for physical oceanography, and I recommend publication in Ocean Science. I do have a few remarks and questions which are listed below.

>>> Thank you for reading our article and recommending it for publication

Main remarks

- At one point I was a bit confused about the focus of the paper: is it about IT dynamics in the Amazon shelf? Is it about how we can study the IT signal in SWOT data? Is it about a new tidal prediction model (ITkar)? I think the paper would gain in visibility and clarity if this was made more explicit early in the manuscript – starting with the abstract. Also, as far as the tidal prediction model is concerned, you could more clearly discuss how you method compare with the one proposed by Egbert and Erofeeva (GRL 2021), which seems quite similar to me.

>>> The objectives of this study are twofold: first to study the internal tide patterns from SWOT data, and second, to estimate a new tidal model from these data to removes as much of the internal tide signal as possible. We have rewritten the introduction and abstract and reorganised the paper to highlight each of them. We hope you find these changes useful.

We haven't discussed the differences between the two methods, as Egbert and Erofeeva (2021) focuses more on the incoherent internal tide (L260 to L262). They do not provide an internal tide model against which we can compare our results.

- The abstract ends with "PCA also gives an overview of the daily variability of ITs", but I do not really see where this is discussed in the manuscript.

    >>> This part has been removed from the abstract.

- The introduction is quite long, but it is relevant and quite well written. Regarding the paragraph (l.~42-51), which is about IT mapping from conventional altimetry: I think that, in the context of your work, some studies on internal tide extraction (and wave/balanced flow disentanglement) from SWOT data could be discussed here (e.g. Egbert & Erofeeva GRL 2021, Le Guillou et al JAMES 2021, Wang et al GRL 2022). This is particularly relevant if the development of an IT prediction model is considered central to your paper (see my first comment on the subject of the manuscript).

>>> Following your recommendations, we have mentioned these studies between L54 and L56 of the new version of the paper.

Minor comments

- I would suggest trying to enlarge the figures that display tracks for better readability.

>>> OK, I will try to make the figures as visible as possible in the final version

- l.15: "the internal tide" or "internal tides".

>>> internal tides

- l.18: You use past tense here, but present tense is used throughout abstract, so I would suggest switching to present tense.

>>> Following your recommendation, we use the present tense

- l.61: "harmonic analysis will only retain local amplitude and phase locked contributions": this is not very clear to me, what does local refer to? Point-by-point in space?

>>> Yes, local means point-to-point in space. But I've removed this part of the sentence as it does not give worthful information- l.68: "get insight of" -> "get (some) insight(s) into"

>>> The sentence has been modified ( L72 to L77)

- l.88-110: I found this presentation of the dynamics in the region very relevant

>>> Thank you

- l.117: "we'll" -> "we shall" or "we will"

>> we will

- Figure 1, sentence l.115-117 and l.143: I think it could be helpful to show the mesoscale currents (e.g. time-averaged) on the figure – maybe duplicate the figure to not overload it, or at least to give more information on the typical RMS of the mesoscale currents (e.g. from DUACS) for comparison with the IT SLA magnitude (l.143).

>>> Figure 1 has been revised and we have added the snapshot of the Duacs product.

- l.122: dot is missing at the end of the sentence

>>> The paragraph has been modified (L124 to L128 ) and a dot added at the end.

- l.124: "eastern" -> "western"

>>> Thank you, the word has been modified

- l.126: very minor comment, but why are the letters in this non-intuitive order?

>>> The order is the same as in Tchilibou et al. (2022), who kept sites A and B as defined in Magalhães et al. (2016) and added the other sites.

- Section 1 or 2: the HRET product is not described: you could consider giving a short description of the methodology, the input data and the contents of the dataset, as well as the source of the dataset besides the reference to the JPO paper (https://ingria.ceoas.oregonstate.edu/fossil/HRET). Please also specify the version used.

>>> We've added a paragraph at the end of sub-section 1.1 to describe HRET.

- l.131 "in netcdf or zcoll format": this information does not seem very relevant to me

>>>This information has been removed.

- l.132: "variable available in zcoll" -> "naming convention used in the dataset"

>>> Thank you, the text has been changed (L139).

- l.155 and below, Fig.2 and Fig. 3: you might consider further describing how the FFT filtering (and spectra) is done: is it only along the along-track dimension? Do you average in the cross-track dimension for the spectra shown in Fig 3? Do you use windowing?

>>> We have added details on spectral analysis (L169-172) and filtering (L182). The 2D spectra are computed in the time and along track dimensions and then averaged over the cross-track dimension. The tukey 0.25 window is used for windowing.

- l.183: If I recall correctly, Solano et al (2023) discuss interferences between mode 1 and mode 2 to explain patterns of surface HKE, but not for the generation of higher modes. However, it is known that this site hosts strong non-linear internal waves (which is the focus of Solano et al), which are associated with harmonics in time and in space, associated with a deviation from sinusoidal shape in the signals. So perhaps this could explain why you see a "high mode" content here (as interpreted from the horizontal scale, not the vertical structure)?

>>> The reference has been removed and the term ''interference'' replaced by ''desintegration''.

- l.197: I know it is common to assume that harmonic analysis is a standard procedure in the physical oceanography community, but there are actually different methods (FFT-based, complex demodulation, least-square fitting), so I would suggest to briefly mention what is the method employed here

>>> We use the method based on least-squares fitting, the precision was done as recommended (L214).

- l.206: which modes from HRET are included?

>>> HRETv8.1 includes low modes, but mode 2 is negligible in our study area. The precision has been done at L154 and repeated at L163.

- l.215: Could this variability also be related to non-tidal motions?

>>> It is possible that some of the variability is non-tidal. For example, there may be sub-mesoscale residuals that have not been removed by DUACS.

- l.217 and Fig. 6: to my knowledge, S2 is included in HRET, so why add the corresponding plot in Fig. 6? (Maybe there is no signal in this region?)

>>> There is S2 signal in this region but the estimations are not very accurate (Zaron personal communication)

- l.249: I think that strictly speaking, this is not "prediction" but rather "extraction" (or "estimation") – which is corroborated by the following paragraph.

>>> This paragraph and the whole of section 2.2 have been rewritten and moved to section 4. In particular, the use of M2 atlases to correct SLA is discussed.

- l.253, "The reason for this is not clear to us": could this be because HRET has almost no signal at larger scales by construction, since the extraction method uses spatial polynomial fits which filter the signal, why ITkars does not and therefore has some variability at these larger scales?

>>> Polynomial fitting may explain why the large-scale signal is not affected by HRET. ITkars does not use a priori spatial polynomials or wave patterns for fitting which may explain why it has some variability at larger scales.

Sentence has been removed and replaced by: ": HRET has almost no signal at large scale by construction, while ITkars can capture some variability at large scales due to short time-series and no fitting approximation". cf L370

- l.296, also l.460 and 475: the IT signal has multiple frequencies (so the coherent part is a combination of several cyclostationary signals): do you have any idea of what is the implication of this on the behavior of PCAs?

>>> We think that the implication is that the PCA produces, even for the main modes, a linear combination of the different frequencies, hence the use of the harmonic analysis to isolate them.

- l.297: It would be interesting to discuss why PCA is expected to improve the results. My guess is that it exploits spatial correlations in the IT field (since it extracts spatially correlated structure), which helps because the coherent IT signal is highly correlated in space.

>>> Yes. Coherent internal tides have the strongest spatial correlations. PCA is particularly effective at identifying and extracting these spatial correlations, thus separating coherent internal tides from incoherent internal tides. We have added a sentence to summarise this argument (L257).

- l.299: Could you confirm that the PCA is performed on raw snapshots?

>>> PCA is performed on the SLA defined in equation 1.

- l.300: sentence starting with "at each point": I was confused at first because I thought normalisation was performed independently at each point, but I guess that what you did is $\sum_t\sum_{pixel} y_{t,pixel}^2 = 1$, right?

>>> We performed the normalisation independently at each point.

**- Fig. 8: There are some discontinuities in the time series that are very surprising. Do you know why?**

>>> The discontinuities are certainly a reflection of the lack of SWOT data for certain cycles.

**- Fig. 8: I am a bit confused: principal components are the spatial patterns (modes), while the time series correspond to the projection of the data on the set of modes, right?**

>>> Spatial patterns extracted by PCA are often referred to as 'modes'. We have chosen not to use this terminology to avoid confusion with baroclinic modes. The time series associated with the spatial modes are called principal components (PCs). However, it is possible that some authors refer to the spatial patterns as PCs.  In this case the time series is called score or coefficient.

**- l.339 and l. 345: "dotted" -> "dashed"**

>>> Ok, dotted has been replaced by dashed

**- l.359: I do not see this on the Figure and I am not sure to understand what you mean here**

>>> We wanted to say that Figure 11d (now Figure 10d)  shows that the ITkars M2 model also contains small scale structure t

**- Fig. 11: I would suggest putting the colour bars inside the frames and adding HRET for comparison, here**

>>> The figure has been redrawn with HRET and ITkars added for easier comparison.

**- l.421: "and finally." -> strange**
>>> It was a mistake. Word withdrawn

**- l.426: it might be interesting to have some quantitative metrics to support the meaning of "close" (e.g., computing RMSE between different estimates?)**

>>> We have replaced ''close'' with "agree". We thought it would be more interesting to compare the reductions variance and spectra as in section 4.  This is because we are making tidal predictions that take into account both amplitudes and phases.

**- l.462, sentence beginning with "Another": I may be misunderstanding, but it seems to me that this is the same mechanism as what was said just before (so it is not "Another possibility", but rather "an interpretation for this").**

>>> We think that the two mechanisms are possible and different. The first because of the time lag between PCA1 and PCA2 and the second because of the intensification in area 2. We are continuing our analysis to better understand what each of these components corresponds to.

Sentence has been changed by : ". Instead of being the real and imaginary parts of a signal, PCA1 and PCA2 could represent the same phenomenon and highlight its evolution in area 2 in the middle of the swaths: the moderation of internal tides in area 2 with PCA1 and their intensification with PCA2."  cf L494

**- l.470: again, I did not really see the daily variability being addressed in the paper**

>>> You are right, it is not a result. We wanted to highlight the fact that the PCA, through the principal components, makes it possible to explore the daily variability of the amplitude of the internal tide, more as a perspective of this work.

Sentence has been changed by: "The principal components (time series) of PCA1 and PCA2 provide an overview and an opportunity to study the daily variability of internal tide amplitude, a perspective difficult to access with conventional altimetry missions." cf L502

---

## Referee Report (RR1)

I am grateful to the authors for carefully responding to all my remarks and comments, and for making some changes to the manuscript which, in my opinion, have resulted in some improvements and clarifications. I recommend the manuscript for publication in Ocean Science, although I have one final comment that should be addressed.

This comment concerns the implementation of the PCA and follows to the authors' response to my previous comment on this issue (l.300 in the first version of the manuscript). I know understand that the Sea Level Anomaly is pointwise normalised by the standard deviation (along the time coordinate) prior to computing the covariance matrix to obtain the PCA modes and eigenvalues. That is, if we call $\eta(x,t)$ (dropping $y$ here for simplicity) the initial sea level time series, and $\eta' = \eta - \langle\eta\rangle_t$ (where the brackets denote averaging over the variable denoted by the index), the PCA is computed on a "reduced" variable $\tilde{\eta}(x,t) = \eta'(x,t)/\sigma(x)$ where $\sigma = \sqrt{\langle\eta'^2\rangle_t}$. Although this does not seem to be critical for the data analysed here, probably because the spatial variations of the time variance are moderate, to my knowledge, this is not a standard procedure and needs to be justified (if it is a documented procedure, please add a reference). As a result, the PCA modes do not account for the variance of the SLA ($\eta'$) but of the reduced variable $\tilde{\eta}$ (which is 1), and the PCA modes (and eigenvalues) are different that what would be obtained by performing a PCA analysis on $\eta'$ (and they are not simply related by a factor $\sigma^2$). As far as I understand, the authors then project the SLA $\eta'$ onto the modes for the reduced SLA, which is not a standard procedure either. Indeed, while the modes form an orthonormal basis, implying that one could use either basis of modes – PCA($\eta'$) or PCA($\tilde{\eta}$) –, one loses the properties that the truncated series over the PCA modes is optimal in terms of variance captured. So it would be surprising if your procedure gave better results than using PCA on $\eta'$.

All together, I think a clarification and justification of the procedure should be given in the manuscript.

---

## Author Response (AR2)

I am grateful to the authors for carefully responding to all my remarks and comments, and for making some changes to the manuscript which, in my opinion, have resulted in some improvements and clarifications. I recommend the manuscript for publication in Ocean Science, although I have one final comment that should be addressed.

>> Thank you for reviewing our manuscript and recommending it for publication.

This comment concerns the implementation of the PCA and follows to the authors' response to my previous comment on this issue (l.300 in the first version of the manuscript). I know understand that the Sea Level Anomaly is pointwise normalised by the standard deviation (along the time coordinate) prior to computing the covariance matrix to obtain the PCA modes and eigenvalues. That is, if we call $\eta(x,t)$ (dropping $y$ here for simplicity) the initial sea level time series, and $\eta' = \eta - \langle\eta\rangle t$ (where the brackets denote averaging over the variable denoted by the index), the PCA is computed on a "reduced" variable $\tilde{\eta}(x,t) = \eta'(x,t)/\sigma(x)$ where $\sigma = \sqrt{\langle\eta'2\rangle}t$. Although this does not seem to be critical for the data analysed here, probably because the spatial variations of the time variance are moderate, to my knowledge, this is not a standard procedure and needs to be justified (if it is a documented procedure, please add a reference). As a result, the PCA modes do not account for the variance of the SLA ($\eta'$) but of the reduced variable $\tilde{\eta}$ (which is 1), and the PCA modes (and eigenvalues) are different that what would be obtained by performing a PCA analysis on $\eta'$ (and they are not simply related by a factor $\sigma2$). As far as I understand, the authors then project the SLA $\eta'$ onto the modes for the reduced SLA, which is not a standard procedure either. Indeed, while the modes form an orthonormal basis, implying that one could use either basis of modesPCA($\eta'$) or PCA($\tilde{\eta}$)−, one loses the properties that the truncated series over the PCA modes is optimal in terms of variance captured. So it would be surprising if your procedure gave better results than using PCA on $\eta'$. All together, I think a clarification and justification of the procedure should be given in the manuscript.

>> We understand your question. Point-by-point standardisation is another way of doing standardization (Jolliffe and Cadima, 2016). In the context of PCA, it makes it possible to limit the effects of large-scale variation and structure. However, in response to your concern, we have restarted the PCA analysis using standard normalization. The impact on our results is negligible, as we have already removed the large scale by using DUACS.

The sentence ''then normalized the time series to ensure that the time mean, and the standard deviation become zero and one respectively'' has been changed by: **then normalized the SLA to ensure that the global mean and standard deviation become zero and one respectively (L264-265).**